



# An overview of the diurnal cycle of the atmospheric boundary layer during the West African monsoon season: Results from the 2016 observational campaign

Norbert Kalthoff[1], Fabienne Lohou[2], Barbara Brooks[3], Gbenga Jegede[4], Bianca Adler[1], Karmen Babić[1],
Cheikh Dione[2], Adewale Ajao[4], Leonard K. Amekudzi[6], Jeffrey N. A. Aryee[6], Muritala Ayoola[4],
Geoffrey Bessardon[3], Sylvester K. Danuor[6], Jan Handwerker[1], Martin Kohler[1], Marie Lothon[2], Xabier
Pedruzo-Bagazgoitia[5], Victoria Smith[3], Lukman Sunmonu[4], Andreas Wieser[1], Andreas H. Fink[1], and
Peter Knippertz[1]

[1]Institute of Meteorology and Climate Research, Karlsruhe Institute of Technology (KIT), Germany
[2]Laboratoire d'Aérologie, Université de Toulouse, CNRS, UPS, France
[3]National Centre for Atmospheric Science, School of Earth and Environment, University of Leeds, United Kingdom
[4]Department of Physics & Engineering Physics, Obafemi Awolowo University, Nigeria
[5]Wageningen University and Research, The Netherlands
[6]Department of Physics, Kwame Nkrumah University of Science and Technology, Kumasi, Ghana

*Correspondence to*: Norbert Kalthoff (norbert.kalthoff@kit.edu)

**Abstract.** A ground-based field campaign was conducted in southern West Africa from mid June to the end of July 2016
within the framework of the Dynamics-Aerosol-Chemistry-Cloud Interactions in West Africa (DACCIWA) project. It aimed
to provide a high-quality comprehensive data set for process studies, in particular into interactions between low-level clouds
(LLCs) and boundary-layer conditions. In this region missing observations are still a major issue.  During the campaign,
extensive remote sensing and in-situ measurements were conducted at three supersites: Kumasi (Ghana), Savè (Benin) and
Ile-Ife (Nigeria). Daily radiosoundings were performed at 0600 UTC and 15 intensive observation periods (IOPs) were
performed during which additional radiosondes were launched every 1.5 to 3 hours. Remotely piloted aerial systems were
also operated during the IOPs. Extended stratiform LLCs form frequently in southern West Africa during the night time and
persist long into the following day. They affect the radiation budget and hence the evolution of the atmospheric boundary
layer and regional climate. The relevant parameters and processes governing the formation and dissolution of the LLCs are
still not fully understood. This paper gives an overview of the diurnal cycles of the energy-balance components, near-surface
temperature, humidity, wind speed and direction as well as the conditions (LLCs, low-level jet) in the boundary layer at the
supersites and relates them to synoptic-scale conditions (monsoon layer, Harmattan layer, African easterly jet, tropospheric
stratification) in the DACCIWA operational area. The characteristics of LLCs vary considerably from day to day, including
a few almost cloud-free nights. During cloudy nights we found large differences in the LLC's formation and dissolution
times as well as in the cloud-base height. The differences exist at individual sites and also between the sites. The synoptic
conditions are characterised by a monsoon layer with south-westerly wind, on average about 1.9 km deep, and easterly wind
above; the depth and strength of the monsoon flow show great day-to-day variability. Within the monsoon layer, a nocturnal
low-level jet forms in approximately the same layer as the LLC. Its strength and duration is highly variable from night to
night. This unique data set will allow us to test some new hypotheses about the processes involved in the development of
LLCs and their interaction with the boundary layer and can also be used for model evaluation.

Keywords: Low-level jet, low-level stratus clouds, nocturnal boundary layer, West Africa




# 1 Introduction

During the West African summer monsoon season, stratiform low-level clouds (LLCs), with typical cloud-base heights of only a few hundred metres above ground, frequently form over southern West Africa during nights that lack deep convection (Schrage et al., 2007; Knippertz et al., 2011; Schrage and Fink, 2012). According to van der Linden et al. (2015), their extent
covers an area of about 800 000 km². As the clouds often persist long into the following day, they control the daytime radiative energy supply at the Earth's surface and, hence, influence the diurnal cycle of the atmospheric boundary layer (ABL) and, by this, they considerably affect the regional climate (Knippertz et al., 2011; Hannak et al., 2017). Up to now, high-quality data in this region are rare and only a few model investigations have been performed to investigate the temporal evolution and spatial distribution of LLCs in this region (e.g. Schuster et al. 2013; Adler et al., 2017). Based on these
investigations, several processes and conditions are expected to be relevant for the formation of LLCs including large-scale advection, orographic lifting, lifting related to gravity waves, latent heat release and vertical mixing of moisture due to shear-generated turbulence underneath the nocturnal low-level jet (LLJ). At Nangatchori in central Benin, Schrage and Fink (2012) found that about 1/3 of the precipitation-free nights were also LLC-free, despite the development of a LLJ. Schrage et al. (2007) concluded that cloudy nights at Parakou (central Benin) were found to occur in flow regimes with a strong
southwesterly monsoon flow in the Guineo-Soudanian zone, a weak African Easterly Jet (AEJ) and an enhanced Tropical Easterly Jet (TEJ), while the reverse was the case for the clear nights. However, the lack of understanding of LLC formation has hitherto impeded any accepted explanation of the impact of large-scale monsoon circulation on observed synoptic variations in LLC extent in southern West Africa. Depending on the position of the AEJ during the season, shear between the lower part of the AEJ and the upper part of the usually south-westerly LLJ can also induce turbulent mixing between the
moist monsoon air and dry Harmattan layer above. The LLJ, which is linked to the north–south pressure gradient associated with the Saharan heat low (Parker et al., 2005; Lothon et al., 2008; Abdou et al., 2010), exists frequently in this region and is found approximately in the same layer as the LLCs. It is unclear which role middle and upper-level clouds play for the nocturnal cloud formation and their morning dissolution as well as for the growth of the cloud-topped daytime boundary layer on the following day (Leung et al., 2016). Aerosols, mainly emitted from the urban agglomerations along the West
African coast, and their north-eastward transport with the monsoon flow are also suspected to impact on cloud characteristics in southern West Africa (Knippertz et al., 2015a). The previous studies suggest that the relevance of the various processes differs spatially but their respective contributions are neither fully understood nor verified by means of observations.

To address these issues, a concerted measurement campaign within the framework of the Dynamics-Aerosol-Chemistry-Cloud Interactions in West Africa (DACCIWA) project was conducted in summer 2016. The overall goal of DACCIWA is
to significantly advance the understanding as well as to improve the capability to monitor and realistically model key interactions between surface-based emissions, atmospheric dynamics and chemistry, clouds, aerosols, and the climate over West Africa (Knippertz et al., 2015b). To achieve these goals, measurements at three supersites: Kumasi (Ghana), Savè (Benin), and Ile-Ife (Nigeria) (see Figs. 1a, b), were conducted and coordinated with airborne measurements of three research aircraft (Flamant et al., 2017). The observations were complemented by additional radiosoundings from existing
national and reactivated African Monsoon Multidisciplinary Analysis (AMMA) networks (Parker et al., 2008). A detailed description of the field activities including details of the radiosonde campaign in summer 2016 is given by Flamant et al. (2017). The strategy for the ground-based measurements was to focus on the processes expected to be involved in LLC formation by applying a synergetic use of in situ and remote sensing observations. In situ measurements cover the near-surface meteorological conditions and the radiation and energy balance components at the Earth's surface. Remote sensing
and in situ observations provide highly resolved thermodynamic conditions in the first kilometres and radiosondes provide profiles of the whole troposphere. Besides radiosondes, which provide base and summit of LLC at given times, remote



sensing systems allowed continuous monitoring of various cloud characteristics (cloud onset, -base, -top, and –base fraction). Brooks et al. (2017) provides a detailed report about these measurements at the ground-based supersites and the available data. A description of the synoptic conditions that prevailed during the period of the DACCIWA campaign is given in Knippertz et al. (2017). In this paper, we give an overview of the meteorological conditions at the three ground-based

supersites. In particular, we aim (i) to provide a characterisation of the local conditions, (ii) to investigate spatial and temporal variability, (iii) to demonstrate the potential of the data set for investigating the dependence of the LLC formation on the synoptic and mesoscale conditions and (iv) to provide some guidance for modellers. The paper is organised as follows. Section 2 describes the measurement sites and data used in this overview study. Section 3 surveys the near-surface and tropospheric conditions for the whole measurement period. Section 4 presents average diurnal cycles of energy-balance

components, near-surface meteorological variables and boundary-layer conditions, and Section 5 summarises and concludes the main findings.

**2 Measurement sites and available observations**

The DACCIWA ground-based measurement campaign lasted from 14 June to 30 July 2016 and encompassed the three

weeks of the airborne campaign (27 June to 16 July 2016), during which three European aircraft conducted research flights across Ivory Coast, Ghana, Togo and Benin (Flamant et al., 2017). During the ground-based campaign, a comprehensive set of instruments was deployed at the three DACCIWA supersites (Figs. 1a, b), namely near the cities of Kumasi, Savè, and Ile-Ife. The three stations are characterised by more (Kumasi, Ile-Ife) or less (Savè) hilly terrain but no significant mountains (Figs. 1c, d, e). The ground campaign consisted of continuous in situ and remote sensing observations as well as intensive

observation periods, during which additional measurements were performed (mainly consisting of frequent radiosonde releases and flights from remotely piloted aircraft systems). In total, 15 IOPs were conducted. An overview of the complete set of instrumentation and measurements at the supersites is given by Brooks et al. (2017). Here we restrict the description of instruments and measurements to those, which allow surveying the meteorological conditions in the whole investigation area and thus spatial differences: that is, to observations that are available at two of the supersites at least. These are

measurements of: (i) the average near-surface meteorological parameters, (ii) the conditions in the atmospheric boundary layer, including cloud characteristics, and (iii) the thermodynamics and dynamics in the whole troposphere. The instrumentation deployed at the different supersites and used in this paper is provided in Table 1b. The measurement heights of the near-surface observations and derived quantities (temperature, $T$, specific humidity, $q$, wind speed, $v$, wind direction, $WD$), radiation and energy balance components (shortwave up- and downward radiation, $R$ and $G$, respectively, longwave

up- and downward radiation, $L\uparrow$ and $L\downarrow$, respectively, total net radiation, $Q_0$, sensible heat flux, $H_0$, latent heat flux, $LE_0$, soil heat flux, $B_0$, turbulent kinetic energy, $TKE$, flux Richardson number, $Rf$) for the three supersites are listed in Table 1a. Note that the amount of data used for the composites of radiation and energy balance differs for the three sites due to instrument failure or quality-flagged periods, but for most of the quantities the data availability is > 80%. The turbulent fluxes for the Savè site are calculated with the TK311 software (Mauder et al., 2013), for Kumasi site according to Aubinet et al. (2012)

and for Ile-Ife a customized eddy covariance program, which runs under the Campbell Scientific software Loggernet, was used.

To monitor the thermodynamics and dynamics of the atmospheric boundary layer, continuously running active (Ultra High frequency (UHF) wind profiler in Savè, sodars in Kumasi and Ile-Ife) and passive (microwave radiometers in Kumasi and Savè) remote sensing systems were operated. Information on integrated water vapour ($IWV$) and liquid water path ($LWP$)

from the radiometers is obtained with a retrieval algorithm provided by the University of Cologne (Löhnert and Crewell,




2003; Löhnert et al., 2009). We trained the algorithm on a set of more than 12,000 radiosonde profiles measured at Abidjan, Ivory Coast, between 1980 and 2014.

Additionally, radiosoundings were performed at 0600 UTC synoptic time. As it is convention that the radiosondes should be close to the tropopause at the nominal time, radiosondes were launched at 0500 UTC at Savè and mainly between 0530 and 0600 UTC in Kumasi. This synoptic time was chosen because the LLC cover was expected to be most intense in the morning hours. The radiosounding data are used to characterise the monsoon and Harmattan flows, the AEJ, the LLJ, to determine the tropospheric stratification and to obtain cloud base and cloud top. The estimation of cloud-base and cloud-top height is achieved by applying the criteria of Wang and Rossow (1995), and is based on relative humidity, $RH$. The low-level clouds are defined as the lowest cloud layer, which fulfils the following three criteria: (i) The cloud layer is the layer where relative humidity is larger than 99%. (ii) The cloud-layer depth must be larger than 100 m (this avoids misclassification of thin fog layers in Kumasi during 4 nights). (iii) The cloud-top height is the level where $RH$ equals 99% and $RH$ shows a decrease of 3 % in the 100-m layer above and with an at least 300 m deep layer of less than 99% of $RH$ above.

The temporal evolution of cloud characteristics, i.e. cloud occurrence, -base, and –cover, is obtained by ceilometers in Kumasi and Savè as well as net infrared radiation available at all three sites. Net longwave radiation showed to be a good proxy for LLC occurrence. As ceilometers from different manufacturers (Lufft in Savè and Campbell Scientific in Kumasi) were used, we expect discrepancies in the derived cloud-base height due to differences in the attenuated backscatter coefficient profiles and different manufacturer algorithms to estimate cloud-base heights from the profiles (a comparison of both ceilometers is described in Madonna et al., 2015). The detected number and temporal resolution of cloud-base heights provided from the different manufacturer algorithms differ (for manufacturers see Table 1b): In Savè, up to three cloud base heights are output every minute, while in Kumasi, up to five cloud-base heights are reported every 10 seconds. In order to increase the comparability for cloud-base height at both sites, we average cloud-base heights at Kumasi over 1-min intervals. For this purpose, we group the cloud-base heights in the 1-min intervals into 100 m vertical bins and assign the median value in the bin with the most number of cloud-base heights to the new 1-min cloud-base height.

## 3 Near-surface and atmospheric conditions for the whole campaign

The 0600-UTC radiosoundings from Kumasi and Savè are used to generate 7-week mean profiles and time series of wind speed and direction (Fig. 2) as well as temperature and humidity (Fig. 3). These diagrams also indicate different phases of the monsoon season, which were distinguished by Knippertz et al. (2017) mainly based on the north-south precipitation difference between the coastal and the Soudanian-Sahelian zones. These are: the pre-onset phase characterized by a rainfall maximum near the coast (before 21 June, phase 1); the post-onset phase during which the rainfall maximum occurred inland (22 June - 20 July, phase 2); the wet westerly regime when the rainfall maximum shifted back to the coast (21 – 26 July, phase 3); and the recovery of the monsoon with a shift of the rainfall maximum inland (27 July until the end of the campaign, phase 4). A specific period within phase 2 is indicated "vortex", during which an unusual development occurred (09 – 16 July): In the north, a cyclonic feature slowly propagated from eastern Mali to Cape Verde and in the south, an anticyclonic vortex tracked in the west-northwesterly direction along the Guinean coast (see Knippertz et al. 2017 for a more detailed description).

### 3.1 Mean profiles

Based on the mean wind profiles, we determine the average depth of the monsoon layer (Fig. 2, left). At both Kumasi and Savè, a distinct minimum in mean wind speed ($\leq 1.5$ m s$^{-1}$) and a shift in wind direction from southwest to east occur at





about 1.9 km above ground level (a.g.l.). This wind shear zone is often used to define the height of the monsoon layer (e.g. Fink et al., 2017). This height shows great variability with time as indicated by the increase of the standard deviation of the wind direction with height. The standard deviation of the wind direction is calculated according to Yamartino (1984). In the monsoon layer, a mean wind speed maximum of around 6 m s$^{-1}$ at Savè and 8 m s$^{-1}$ at Kumasi occurs at about 400 m a.g.l.

This wind maximum is related to the LLJ that is usually still present at 0600 UTC. Possible reasons for the greater mean wind-speed maximum in Kumasi compared to Savè will be investigated in Section 4 in more detail. No distinct AEJ is visible in the mean wind profiles. This is because the AEJ was further to the north in the second half of the campaign and due to averaging over opposing winds, i.e. normally winds are easterly but westerly during the vortex period (Fig. 2, right). The standard deviation of the wind speed, which is more than ± 3 m s$^{-1}$ in most of the layers at both sites, reflects strong day-

to-day variations.

In the monsoon layer, the mean *RH* varies between 80 and 100 % at both sites (Fig. 3, left). Above the monsoon layer, *RH* is about 75 % at Kumasi. At Savè, *RH* has a minimum of 75 % at about 3 km a.g.l. and increases to about 85 % above. That means, on average the atmosphere between 3 and 5 km a.g.l. at Kumasi is somewhat drier than Savè. This could be caused by convective activity upstream of Savè in Nigeria. Note that the standard deviation of *RH*, especially above 1 km, is quite

high (more than ± 10 %). The mean temperature profiles are rather similar at both sites, but the atmosphere at Savè is slightly warmer than at Kumasi. The mean gradient of the potential temperature is about 6 K km$^{-1}$ between 0.5 and 2 km layer and about 5 K km$^{-1}$ between 2 and 5 km. As at the same time the mean gradient of the saturated equivalent potential temperature in this two layers is negative (not shown), the lower troposphere is conditionally unstable. At both sites, the standard deviation of the potential temperature is on the order of ± 1 K.

**3.2 Day-to-day variability**

The large standard deviation of wind speed and direction can be explained by the atmospheric conditions on individual days (Fig. 2, right). In the Harmattan layer above the monsoon layer, the AEJ is strong at both sites (mainly > 10 m s$^{-1}$) during the first half of the observation period. On 11 July, a cyclonic vortex slowly propagates west-northwestwards from Gabon across the Gulf of Guinea and reaches Sierra Leone on 14 July (Knippertz et al., 2017). This causes large-scale westerly winds up to

about 3 km a.g.l. associated with a weakening and northward shift of the AEJ. This is also visible in the wind profiles at Savè (11–13 July) and Kumasi (11–14 July). After the vortex period, the mid-level easterly wind remains weak in the second half of the measurement period (15–30 July) at both sites. The large standard deviation in the monsoon layer partly arises from days, which are affected by mesoscale convective systems (MCS), e.g. on 19 June in Savè. As the wind profiles at 0600 UTC result from the monsoon flow with the embedded LLJ, variations are caused by large-scale conditions as well as by the

characteristics of the LLJ. For example, a deep layer with high wind speed can be found on 21 June and 11 July at both sites.

The classical concept is that the monsoon layer is associated with southwesterly wind and moist air. That means, in principle both quantities are good to identify it. Using *RH* as an indicator for the moist layer, Fig. 3 shows that the day-to-day variability of the depth of the moist layer in the lower atmosphere varies by a factor of two or more and does not always coincide with the depth of the southwesterly wind layer. On the one hand, periods occur when the moist layer was much

deeper than the monsoon layer. These periods can often be assigned to MCSs passing the sites, accompanied by deep vertical mixing. For example, on 16 July a long-living MCS (associated with cyclonic system "C" in Knippertz et al., 2017) passes Savè and on 26 July another MCS (associated with cyclonic system "J") affects Kumasi. On the other hand, the moist layer is significantly shallower than the monsoon layer during some periods. For example, the moist layer is only some hundred meters deep at Savè during the end of the pre-monsoon phase. During the last part of the post-onset phase, the moist layer at



Savè is somewhat drier than the weeks before. This period is roughly related to the vortex period when dry air was transported northwards.

Cloud-base and cloud-top heights estimated from *RH* profiles are also included in Fig. 3. Days with precipitation during the radiosounding were excluded from the analysis (Kumasi: 11 and 24 July, Savè: 19 June, 20 and 24 July). The results show

more frequent occurrence of LLCs at Kumasi (88 %) than Savè (64 %) at 0600 UTC (purple dots in right panels of Fig. 3). In Savè, the median height for cloud-base is 227 m a.g.l. and for cloud-top is 587 m a.g.l., while in Kumasi, the median heights are 137 m a.g.l. and 692 m a.g.l., respectively. That means on average, LLCs in Kumasi are more often thicker than in Savè in the morning. As for the other parameters, there is evidence for a link between the monsoon phases and LLC occurrence and depth (see also Fig. 17 in Knippertz et al. 2017). During the pre-onset phase, LLCs are rather thin at both

sites and less frequent at Savè. A higher LLC occurrence is observed during the post-onset monsoon phase, whereas dry air advection during the vortex period tends to reduce the LLC depth at Kumasi and even prevents LLC formation at Savè. For the remaining observation period, LLC formation is more sporadic, partly due to the increase of MCS events. Earlier work has already pointed to the fact the LLC forms in nights that lack deep convection, but farther north, cloud-free non-precipitative nights appear to occur more frequently (e.g. Schrage et al., 2007; Schrage and Fink, 2012). More information

on the temporal evolution of cloud characteristics is given in Section 4.

The *IWV*, calculated from radiosoundings, shows higher values at Savè (median of 55 kg m$^{-2}$) than at Kumasi (median of 50 kg m$^{-2}$) (Fig. 4c). This difference can be attributed to higher absolute moisture content at all levels. During the whole measurement period, no clear trends are visible neither for Kumasi nor for Savè. A similar difference between the two sites exists in the *IWV* calculations based on microwave radiometer data (see section 5). In Kumasi, the vortex period is

accompanied by lower *IWV* values (Fig. 4c). This *IWV* decrease from about 50 to 34 kg m$^{-2}$ could be attributed to dry air that is advected with the south-westerly wind (Fig. 2a) from the area of subsidence in the equatorial zone or even the southern hemisphere. Knippertz et al. (2017) report that the intrusion of dry air resulted in *RH* values as low as 10 % in the middle troposphere over Abidjan in Ivory Coast during this vortex phase. Farther east in Savè, the dry-air advection is much less pronounced and does not show up in the *IWV* (Figs. 2b, 4c).

The convection-related parameters convective available convective potential energy (*CAPE*) and convective inhibition (*CIN*) are calculated from the soundings in Kumasi and Savè and shown in Figs. 4a and 4b. As the soundings are performed at 0600 UTC, both quantities are calculated using the most unstable layer for lifting (Doswell III and Rasmussen, 1994), because surface-based parcels are inappropriate for lifting when a nocturnal surface inversion still exists. At both sites, *CAPE* shows a strong day-to-day variability, ranging from about 100 to about 2000 J kg$^{-1}$, with no clear dependence on the

different monsoon phases (Fig. 4a). The median for the whole period for Kumasi is 434 J kg$^{-1}$ and 508 J kg$^{-1}$ for Savè. We attribute the large day-to-day variability of *CAPE* to convective precipitation events, which are typically followed by a decrease of *CAPE* as described by Schwendike et al. (2010) for the Sahel region and Schrage et al. (2006) for the Soudan region. High values of *CIN* (about 50 to 100 J kg$^{-1}$) exist during the pre-onset phase (14–20 June) and at the end of the post-onset phase (about 14–21 July) of the monsoon (Fig. 4b). During these phases, stronger precipitation occurs (Fig. 4d). This

link can be explained by a build-up of latent instability due to an accumulation of warm, moist air in lower layers in the presence of higher *CIN* values. This ultimately enhances the potential for stronger convection with heavy precipitation (e.g. Browning et al., 2007; Khodayar et al., 2010). The lowest *CIN* values (< 50 J kg$^{-1}$) are mainly observed during the post-onset phase, associated with little precipitation (Fig. 4d), typically generated by local convection. Fink et al. (2006) describe different types of rainfall associated with different *CAPE* and *CIN* values and intensities of precipitation for Parakou, which

is about 130 km north of Savè. It is conceivable but left for future studies if the observed variability in *CAPE* and *CIN* and





the related rainfall events fit into the types of rainfall proposed by Fink et al. (2006). The total amount of precipitation over the whole period ranges from 217 mm in Savè and 258 mm in Kumasi to 271 mm in Ile-Ife. It has to be kept in mind that all precipitation data are based on local measurements, which are not necessarily representative for the amount of convective precipitation in the area. However, the temporal distribution of precipitation in Savè shown in Fig. 4d agrees with that of the

Savè X-band radar, which covers a diameter of up to 200 km (not shown).

The energy exchange at the surface is analysed using the evaporative fraction $EF = LE_0 / (LE_0 + H_0)$, i.e. the ratio of the latent heat flux to the sum of latent and sensible heat fluxes or available energy (Fig. 4d). This quantity was calculated from $H_0$ and $LE_0$ values, which were averaged for the time period from 0900 to 1500 UTC, i.e. when the fluxes are sufficiently high. A value of one indicates that all available energy goes into the latent heat flux, a value close to zero means that the

sensible heat flux dominates and at a value of 0.5 energy is equally distributed between sensible and latent heat fluxes. The median of $EF$ is 0.58 in Kumasi, 0.64 in Savè and 0.69 in Ile-Ife, i.e. on average the majority of the available energy is transformed into latent heat flux – less in Kumasi and more in Ile-Ife, although both are grassland sites. Although the $EF$ at the different sites is rather constant over the campaign, some of the changes of $EF$ are related to rain events; for example, the increases of $EF$ in Savè from about 0.5 on 12 July to 0.75 on 13 July and from 0.6 on 19 July to 0.8 on 20 July. In Kumasi, a

precipitation-generated $EF$ increase from 0.4 to 0.7 is observed from 16 to 17 July. A similar behaviour of the evaporative fraction after precipitation events, when the soil is not saturated, is also reported by Kohler et al. (2010) and Lohou et al. (2014) for the AMMA field campaign.

### 4 Diurnal cycles of near-surface and boundary-layer quantities

The survey of the complete campaign reveals that the conditions differ considerably during the various monsoon phases as well as between the three sites. In this section, we present average diurnal cycles of the cloud characteristics followed by parameters considered relevant for LLC formation.

#### 4.1 LLC characteristics

From the 1-min cloud-base heights obtained from ceilometer measurements, we calculate the cloud-base fraction for 30-min
intervals for the lowest 1000 m above ground (Figs. 5a, b). Cloud-base fraction gives information on the amount of cloud bases in a certain time period and height layer, i.e. a cloud-base fraction of 100 % indicates that at all times in the 30-min interval at least one cloud base is detected in the lowest 1000 m a.g.l. Using a threshold of 100% for LLC detection, it is evident that LLCs develop during many nights at both sites. At 0600 UTC, i.e. at the same time when daily radiosoundings were performed, LLCs occur less often in Kumasi (approximately 57%) than in Savè (approximately 64%), which differs

from the finding based on the radiosonde data (LLCs occur more often in Kumasi than in Savè, Sec. 3.2). However, in Kumasi the results are very sensitive to the selected threshold for cloud-base fraction. Applying a threshold of 80%, LLC occurrence increases to approximately 80% in Kumasi while it remains approximately the same in Savè. Nights without LLCs are found between the 14 and 17 June at Savè, and around the 23 and 24 June and 14–16 July at both sites. The last period is roughly associated with the vortex occurrence, during which dry air masses are transported into the investigation

area (see Knippertz et al., 2017). Using a median cloud-base fraction of 100 % as an indicator for the average onset of LLCs, LLCs approximately form at 0000 UTC in Kumasi and 0300 UTC in Savè (Fig. 5b). The onset of LLCs is accompanied by an increase of the median $LWP$ to about 40 g m$^{-2}$ at Savè and to about 50 g m$^{-2}$ in Kumasi (Fig. 5d) and the net longwave radiation, $L_{net} = L\!\downarrow - L\!\uparrow$, increases to about $L_{net} \approx -10$ W m$^{-2}$ at both sites (Fig. 5e). Using a threshold of $L_{net} \approx -10$ W m$^{-2}$ as a



proxy for the existence of LLCs, as indicated from the comparison of net longwave radiation with cloud-base fraction in Kumasi and Savè, LLCs in Ile-Ife already form at around 2100 UTC.

From cloud-base fraction we cannot obtain the information as to whether the cloud bases are all at the same height or distributed over several layers within the 1000 m layer. To get an idea of the vertical distribution of cloud-bases we also

calculate the frequency distribution of cloud-base height by counting the number of days with at least one cloud base within a respective bin (bin size is 10-minute duration and 20 m vertical range, Fig. 5c). As the backscatter coefficient profiles at Kumasi show no realistic values below around 50 m a.g.l. this layer is masked by the black bar in Fig. 5c. While there is some variability in cloud-base height during the campaign, layers particularly favourable for the occurrence of cloud base can be distinguished. In Savè, surprisingly two layers are evident: one is at around 100 m a.g.l. and the other one at around

300 m a.g.l. The reasons for this need detailed investigations. In Kumasi, cloud-base height occurs mainly in a layer around 200 m a.g.l. Radiosonde profiles at Kumasi indicate that cloud base also occur at lower layers (Fig. 3a). Unfortunately, this cannot be verified by ceilometer data due to missing measurements.

At both sites, the cloud base starts to rise at approximately 0700 UTC, i.e. about one hour after sunrise (time of sunrise see Table 1a). During the rising period the cloud fraction remains close to 100 % (Fig. 5b) and fluctuations in cloud-base height

remain small for some time (Fig. 5c), indicating that the clouds are still stratiform. Eventually a transition to convective clouds occurs, indicated by a decrease in cloud-base fraction and a stronger fluctuation of cloud-base height. On average, the transition is approximately 2 hours after the start of the rising period; however, we find a large variability of several hours. Parallel to the cloud-base rising around 0700 UTC, the $LWP$ at Savè reaches maximum values of up to 100–125 g m$^{-2}$ and $L_{net}$ decreases at both sites (Fig. 5d). Around midday, when the convective boundary layer is well developed, the cloud

fraction has decreased (approximately 40 % in Kumasi, 80% in Savè), $L_{net} \approx$ - 40 W m$^{-2}$ and the cloud-base height is at around 800 m a.g.l. In Ile-Ife, $L_{net}$ starts to decrease at 0700 UTC indicating the dissolution of the LLCs at that site, too. Overall, the LLCs at the individual sites show a considerable variability with respect to their time of formation and dissolution, cloud fraction, $LWC$ and cloud-base height. The same finding holds when comparing the LLC conditions between the different supersites. Next, the diurnal cycles of quantities related to the LLC formation are presented.

**4.2 Radiation and energy balance at the surface**

Figure 6 presents composites of the diurnal cycles of the radiation and energy balance components for the three sites. Figure 6a indicates that the radiation balance components are quite similar at three sites (Fig. 6a). The maximum median $G$ reaches up to approximately 550 W m$^{-2}$ with an interquartile range of approximately 250 W m$^{-2}$ at 1300 UTC. This is caused by the considerable day-to-day variability of cloud cover. The $R$ is about 110 W m$^{-2}$, i.e. the albedo at the three supersites is

approximately 0.2, which is a typical value for grasslands (e.g. Oke, 1987). The longwave radiation shows much less variability.

At all three sites, the median $Q_0$ is of similar magnitude of 400–450 W m$^{-2}$ around midday (Fig. 6b). At this time, the median is accompanied by a significant interquartile range of 200 W m$^{-2}$ (Fig. 6b), as is to be expected given the variability in the radiation components. The partitioning of the available energy ($Q_0 - B_0$) between $H_0$ and $LE_0$, however, differs at the three

sites. In Kumasi, the median $LE_0$ is only slightly higher than $H_0$, which is consistent with a median evaporative fraction 0.58 (Sect. 3). In Savè, $LE_0$ clearly dominates $H_0$, e.g. at noontime the median $LE_0$ being about 200 W m$^{-2}$ while the median $H_0$ is about 120 W m$^{-2}$. At Ile-Ife, the median $LE_0$ is also about 200 W m$^{-2}$ and the median $H_0$ is only 100 W m$^{-2}$ at noontime, resulting in the highest evaporative fraction (0.69) of all three sites. The accumulated daily evapotranspiration, $E_0$, derived from the median $LE_0$ values, amounts to 1.3 kg m$^{-2}$ at Kumasi, to 2.1 kg m$^{-2}$ at Savè, and to 1.6 kg m$^{-2}$ at Ile-Ife. The

turbulent fluxes at all three sites during daytime are associated with great interquartile ranges, due to both the strong day-to-





day variability of available energy and because precipitation events modify the evaporative fraction from day to day (Fig. 4d). At all three sites, around sunset median $Q_0$ becomes negative (around - 10 W m$^{-2}$). The absolute value of $H_0$ is low at night (median between $H_0$ = - 8.5 W m$^{-2}$ in Savè and -0.3 W m$^{-2}$ in Ile-Ife), but by contributing to the compensation of the negative $Q_0$, $H_0$ contributes to the development of a stably stratified nocturnal surface layer, the decoupling of the daytime
mixed layer from the surface layer and the development of an LLJ.

As expected for a tropical region during the monsoon season, the near-surface temperature exhibits only a moderate diurnal cycle with a median diurnal temperature range of about 6 °C at the three sites (Fig. 7a). The median temperature maximum of 29 to 30 °C is reached in the early afternoon (about 1500 UTC). The temperature's day-to-day variation is moderate, too, e.g. the interquartile range at 1500 UTC is only about 2 °C. The diurnal amplitude of the median $q$ is small as well, i.e. only
about 1.0 g kg$^{-1}$ in Kumasi, 1.2 g kg$^{-1}$ in Ile-Ife and 1.8 g kg$^{-1}$ in Savè (Fig. 7a). The median $q$ is highest at Savè (about 19 g kg$^{-1}$ between 1400 to 2000 UTC). This might be attributed to the higher evapotranspiration (Fig. 6b). The microwave-derived $IWV$ values (Fig. 7d) do not show a pronounced diurnal cycle, while precipitation reveals a strong time dependence (Fig. 7d). A period with more precipitation occurs between about 1500 and 2300 UTC. This time period is typical for local convective precipitation. The occurrence of local and patchy precipitation in the late afternoon is confirmed by the rain-radar
data at Savè (not shown).

When the surface layer becomes stably stratified around sunset, as indicated by the positive $Rf$ at all sites (Fig. 7c), the near-surface wind speed and the $TKE$ decrease (Figs. 7b, c). This decoupling of the surface layer from the mixed layer allows an LLJ to develop at all three sites (Fig. 8). The temporal cross section of median wind profiles are based on sodar measurements in Kumasi and Ile-Ife, while for Savè UHF wind profiler measurements are available. Affected by local noise
production from a generator, the sodar at Kumasi did not reach high altitudes, especially at night (Fig. 8a). In Savè and Ile-Ife, the LLJ is well established at about 2000 UTC (Figs. 8b, c). In Savè, the jet core can be found between 300 and 500 m a.g.l., the maximum median wind speed is about 8 m s$^{-1}$ with a decrease of wind speed in the second half of the night. In Ile-Ife, the top of the LLJ is not covered by the sodar. Mean hodographs of the wind speed, calculated for the 200–600-m layer, show a clockwise turning of the wind in the course of the night from south-southwesterly at sunset to southwesterly at
sunrise. The evolution of the nocturnal LLJ at the different sites is similar to observations described by Lothon et al. (2010) and Abdou et al. (2010) from AMMA. In the upper and lower shear zones of the LLJ, turbulence is expected (e.g. Banta et al., 2006) and gravity waves, similar to those at the top of drainage flows (Viana et al., 2010), are likely. Both are expected to have an impact on the LLC formation (Adler et al., 2017) and will be investigated in subsequent case studies.

During the night, the median near-surface wind speed and $TKE$ are higher in Kumasi ($v \approx$ 1.5 m s$^{-1}$, $TKE$ up to 0.5 m$^2$ s$^{-2}$)
than at the two other sites ($v \approx$ 1.0 m s$^{-1}$, $TKE$ up to 0.2 m$^2$ s$^{-2}$) (Figs. 7b, c). Due to higher $TKE$, the surface layer in Kumasi is less stably stratified, i.e. the positive values of $Rf$ are lower, than at the two other sites. We assume that differences in LLJ are likely responsible for the differences in the near-surface conditions. A weaker LLJ results in weaker downward mixing (lower $TKE$). To seek for explanation we inspect the large-scale pressure gradient as a driving force for the LLJ and the orography: (i) there is evidence that the meridional pressure gradient is larger in the western part of the investigation area,
i.e. the geostrophic wind has a larger zonal component at Kumasi, (ii) the 700 m high Atakora mountains in Togo (Fig. 1) cause a reduction of wind speed on the downstream side (as evident in simulations by Schuster et al. 2013), which also affects Savè. Shortly after sunrise, when the surface layer becomes unstable again (Fig. 7c), convective vertical mixing starts and momentum is transferred down to the surface again, as indicated by a $TKE$ increase (Fig. 7c), and the LLJ starts to dissolve (Fig. 8). Simultaneously, near the ground the wind speed increases and the wind direction becoming more westerly
(Fig. 7b).





## 5 Summary and conclusions

A unique, high-quality data set was obtained during the DACCIWA ground-based field campaign. This campaign was performed in southern West Africa from mid June to the end of July 2016 with intensive ground-based measurements carried

out at three supersites, namely in Kumasi (Ghana), Savè (Benin) and Ile-Ife (Nigeria). The aim of this study is to provide a comprehensive overview of the conditions related to the development of nocturnal LLCs including their formation and the transition to convective clouds during the day. In order to allow for the comparison of conditions at the different sites, included here are only the observations which have been performed at two of the supersites at least. Two types of analyses are chosen: time series for the entire 7-week period and average diurnal cycles of meteorological parameters. The main

findings are:

- The monsoon-layer depth, determined from the minimum in wind speed of the 0600-UTC radiosoundings, is found to be on average at approximately 1.9 km. Within the monsoon layer, the mean wind direction is southwesterly and the mean wind-speed maximum, which occurs at about 400 m a.g.l., is about 6 m s$^{-1}$ at 0600 UTC in Savè and 8 m s$^{-1}$ in Kumasi. The wind speed profiles are characterized by the monsoon flow and the embedded LLJ, which still

exists around 0600 UTC. In the Harmattan layer, easterly wind prevails with average wind-speed values of 6 to 7 m s$^{-1}$. The day-to-day variability both in the monsoon and in the Harmattan layer is high and mainly caused by the changes in the strength of the LLJ and position of AEJ, respectively. Between 11 and 14 July, even westerly winds are present in the Harmattan layer in conjunction with the passage of an anticyclonic vortex south of the investigation area (Knippertz et al., 2017). The mean $RH$ profiles show high values in the monsoon layer (80 to 95

%) and a minimum at about 3 km a.g.l. Above, the $RH$ on average was higher in Savè than in Kumasi. This is possibly caused by stronger convective activity upstream of Savè in Nigeria, resulting in deeper vertical mixing of humidity. The depths of the moist layer and the monsoon layer do not always coincide, indicating that on some days dry air intrusions from the Harmattan into the monsoon layer and on other days a transport of humid air from the monsoon to the Harmattan layer takes place. This means, deviating from the classical concept of a monsoon layer

with southwesterly wind and moist air, wind and moisture signals can be quite distinct.

- The diurnal cycle of boundary-layer conditions at the three supersites reveals some interesting features. After sunset, which is about 1800 UTC in the investigation area, the surface layer becomes stably stratified being less stable in Kumasi than in Savè and Ile-Ife, as indicated by a lower flux Richardson number. These spatial differences are also reflected in the near-surface $TKE$ and wind-speed values, being higher during the night in Kumasi ($v \approx 1.5$

m s$^{-1}$ and $TKE \approx 0.5$ m$^2$ s$^{-2}$) than at the two other sites ($v \approx 1.0$ m s$^{-1}$ and $TKE \approx 0.2$ m$^2$ s$^{-2}$). After the surface layer becomes stably stratified, an LLJ develops at all three sites nearly every night being strongest between 2100 and 0200 UTC. The LLJ dissipates gradually after sunrise at around 0600 UTC.

- At all three supersites, LLCs develop on most of the nights. The occurrence frequency depends on the instrument and method used for the detection of LLCs. Based on the 0600-UTC radiosounding, LLCs occurred more often in

Kumasi than in Savè, while the ceilometer measurements indicate the opposite. However, the latter is very sensitive to the applied criteria. The differences concerning cloud characteristics (LLC occurrence, cloud-base and cloud-top height) derived from radiosonde and ceilometer observations will be subject of subsequent detailed investigations. Short episodes without LLCs are found at the beginning of the campaign in Savè, around 23–24 June, and around 14–16 July. Using a cloud-base fraction of 100 % estimated from ceilometer measurements and net longwave

radiation of -10 W m$^{-2}$ as an indicator for the average onset of LLCs, LLCs form at approximately 2100 UTC at Ile-



Ife, at 0000 UTC in Kumasi and 0300 UTC in Savè. In Savè, a two-layered structure of the cloud-base height is distinguishable with a maximum of cloud-base occurrence around 100 and 300 m a.g.l.; the reasons for this are currently being investigated. In Kumasi, the cloud base height is mainly distributed around 200 m a.g.l. On average, the cloud base starts to rise around one hour after sunrise, reaching heights around 800 m a.g.l. at noon. During this

rising period, the clouds remain stratiform, eventually breaking up and form convective clouds. The times of LLC onset, breakup and transformation from stratiform to broken clouds vary considerably, all being responsible for the strong day-to-day variability of the energy-balance components.

The measurements provide a unique data set to solve the two great enigmas namely (1) what are the decisive processes and parameters for LLC formation (energy balance at the Earth's surface, LLJ, depth and the strength of the monsoon flow,

Harmattan and AEJ conditions, cold air advection, and the presence of mid- and upper level clouds) and (2) what determines their variability at temporal and spatial scales? To investigate these relationships and possible feedbacks, detailed process analysis will be performed in subsequent studies. Furthermore, this high quality data set is invaluable for model evaluation and can be used to obtain the initial conditions for large eddy simulations dealing with cloud topped boundary layers as well as for reanalysis. After the DACCIWA embargo period, the data of the three supersites will be available on the SEDOO

database (Brooks, 2016; Derrien et al., 2016; Handwerker et al., 2016; Jegede et al., 2016; Kohler et al., 2016; Wieser et al., 2016) for scientists interested in boundary-layer studies in southern West Africa.

### Acknowledgement

The DACCIWA project has received funding from the European Union Seventh Framework Programme (FP7/2007-2013) under grant agreement no. 603502. We also want thank the staffs from NCAS (National Centre for Atmospheric Science),

KIT (Karlsruhe Institute of Technology) and UPS (Université Toulouse) for helping to install the equipment as well as the people from KNUST in Kumasi, INRAB in Savè for allowing the equipment on their ground and NCAS for providing the instrumentation deployed at Kumasi.

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





Table 1: a) Coordinates and station heights in meter above sea level (m a.s.l.) and measurement heights in meter above ground level (m a.g.l.) of the energy balance station's instruments at three supersites. $Q_0$ designates the total net radiation, $R$ the shortwave up- and $G$ the downward radiation, $L\uparrow$ the longwave up- and $L\downarrow$ the downward radiation, $H_0$ the sensible heat flux, $LE_0$ the latent
5  heat flux, $B_0$ the soil heat flux, *TKE* the turbulent kinetic energy, *Rf* the flux Richardson number, $T$ the temperature, $q$ the specific humidity, $v$ the horizontal wind speed and *WD* the wind direction. b) Manufacturers of the systems at the different sites.

a)

|  | Kumasi | Savè | Ile-Ife |
|---|---|---|---|
| Latitude | N 6° 40' 48.56'' | N 8° 00' 03.6" | N 7° 33' 11.52" |
| Longitude | W 1° 33' 37.76'' | E 2° 25' 41.1" | E 4° 33' 26.70" |
| Height (m a.s.l.) | 266 | 166 | 274 |
| 14 June: sunrise, sunset (UTC) | 0551, 1822 | 0533, 1808 | 0525, 1759 |
| 30 July: sunrise, sunset (UTC) | 0600, 1825 | 0542, 1811 | 0534, 1802 |
| Vegetation type | Short grassland | Grass and bushes (waist-deep) | Short grassland |
| $G$, $R$, $L\downarrow$, $L\uparrow$, $Q_0$ (m a.g.l.) | 1.68 | 3 | 1.7 |
| $H_0$, $LE_0$, *TKE*, *Rf* (m a.g.l.) | 3.5 | 4 | 1.7 |
| $B_0$ (m a.g.l.) | -0.08 | -0.02 | -0.02 |
| $T$, $q$ (m a.g.l.) | 2 | 2 | 1.7 |
| $v$, *WD* (m a.g.l.) | 3.5 | 4 | 1.7 |
| *Precipitation* (m a.g.l.) | 1 | 1 | 1 |

b)

| | | | |
|---|---|---|---|
| Ceilometer | Campbell Scientific | Lufft | - |
| Energy balance | Kipp & Zonen, Campbell Scientific, LI-COR | Kipp & Zonen, Gill, LI-COR | Kipp & Zonen, Campbell Scientific, LI-COR |
| Microwave radiometer | Radiometer Physics | Radiometer Physics | - |
| Sodar | Sensor Technik Simach | - | Metek |
| (Tethered) radiosondes | Vaisala | Météomodem | GRAW |
| UHF wind profiler | - | Degreane Horizon | - |





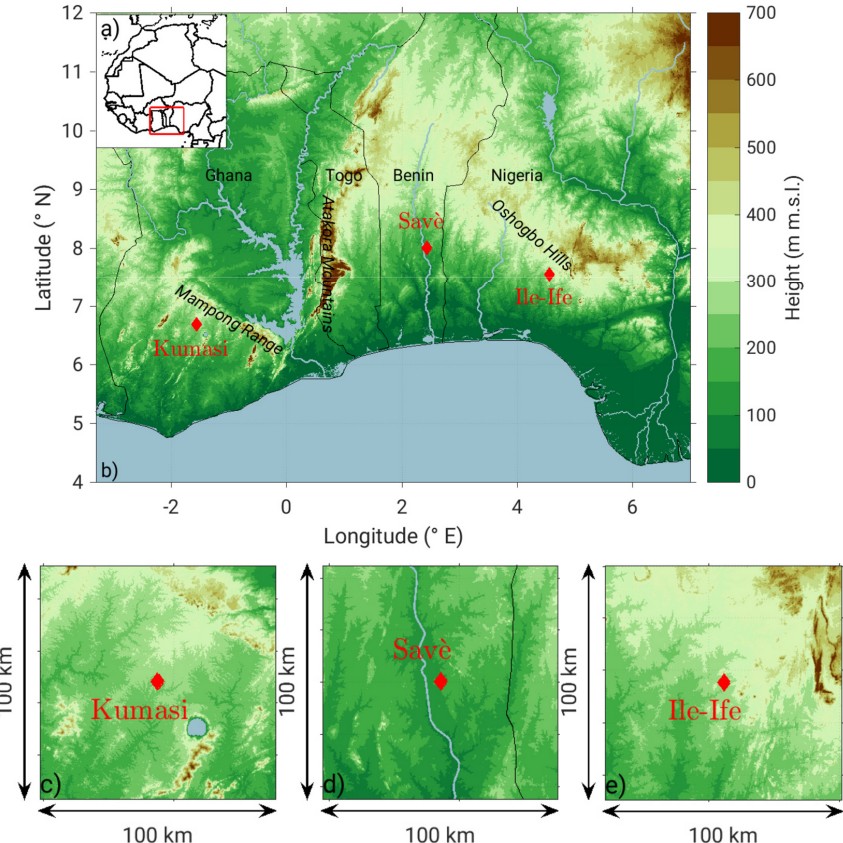

**Figure 1: Location of the DACCIWA investigation area (red box) in West Africa (a) and orography at the three measurement sites Kumasi in Ghana, Savè in Benin, and Ile-Ife in Nigeria (b). More detailed orography in the immediate surrounding of the three supersites Kumasi (c), Savè (d), and Ile-Ife (e). Solid lines indicate country borders with country names given in (b).**





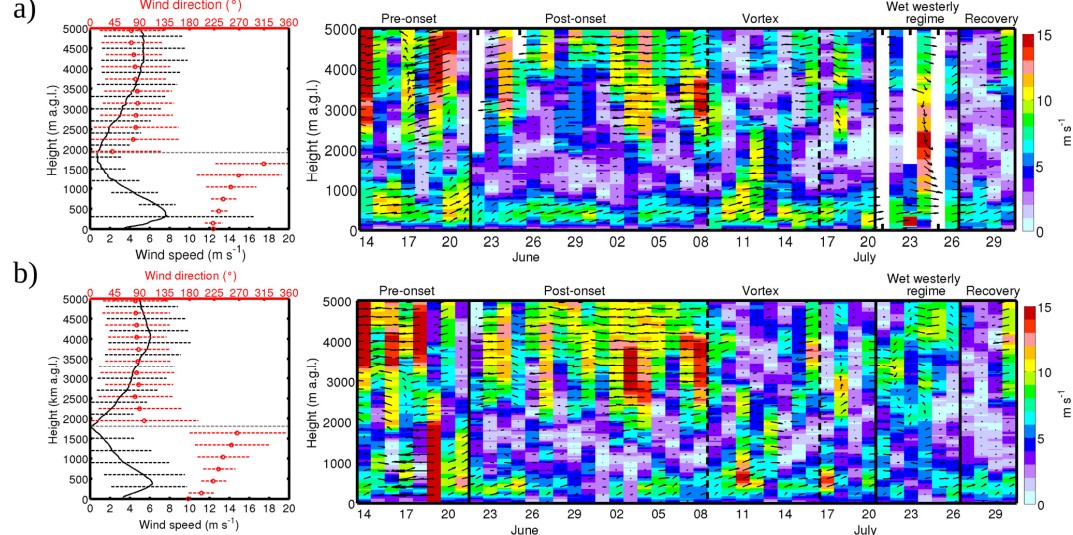

**Figure 2: (left) Mean profiles of wind speed (black solid line) and direction (red dots) plus their standard deviations based on 0600-UTC soundings at Kumasi (a) and Savè (b). The horizontal grey dashed line indicates the monsoon height. (right) Times series of the horizontal wind vector (arrows) and absolute value of the wind speed (colour coded) at 0600 UTC at Kumasi (a) and Savè (b).**
5  **Different phases of the monsoon are indicated according to Knippertz et al. (2017).**





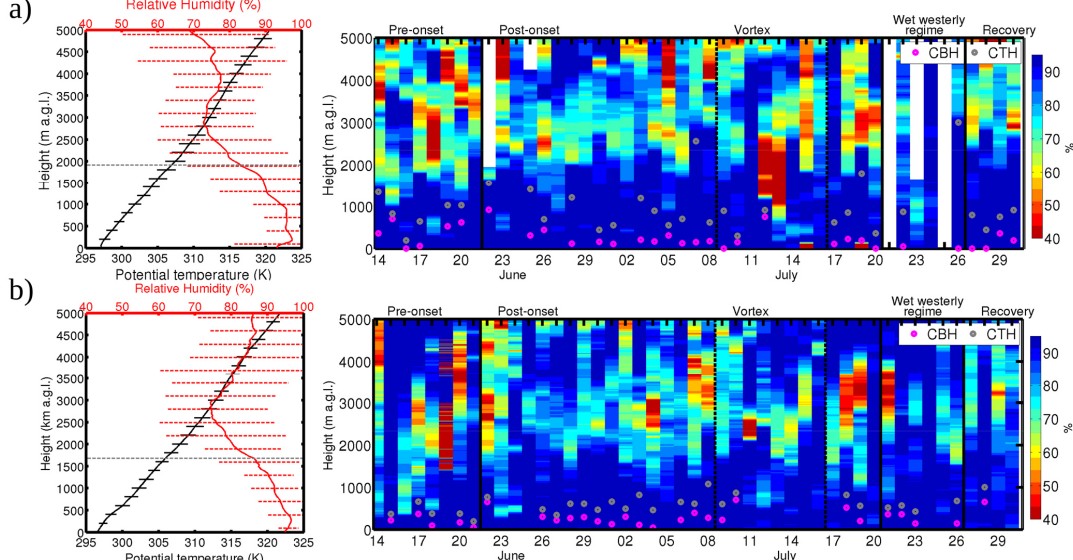

**Figure 3: (left) Mean profiles of relative humidity (red solid line) and potential temperature (black solid line) plus their standard deviations based on 0600-UTC soundings at Kumasi (a) and Savè (b). The horizontal grey dashed line indicates the monsoon height. (right) Times series of relative humidity (colour coded) at 0600 UTC at Kumasi (a) and Savè (b). The pink and grey circles indicate cloud base and top estimated from the relative humidity profiles, respectively. Different phases of the monsoon are indicated according to Knippertz et al. (2017).**

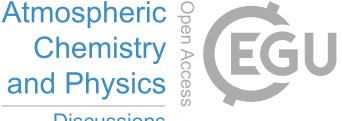



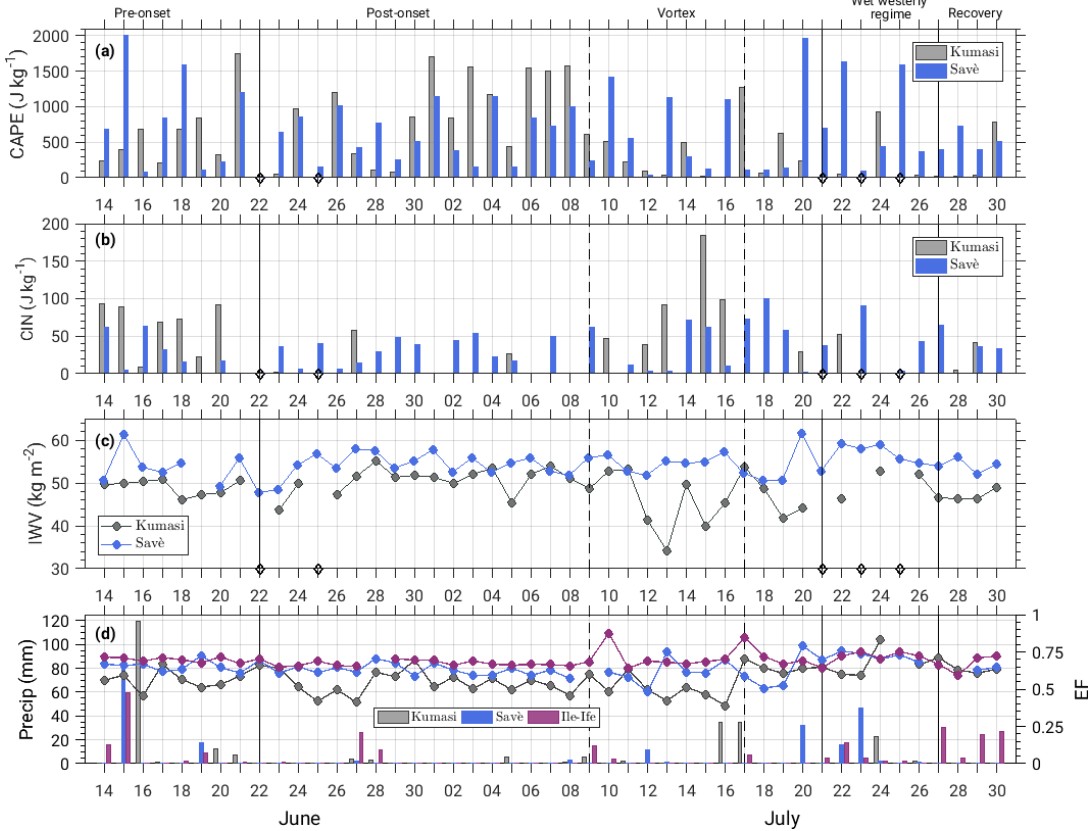

**Figure 4: Time series of *CAPE* (a), *CIN* (b), and *IWV* (c) from the 0600 UTC radiosonde data for Kumasi and Savè. Diurnal precipitation sum and evaporative fraction (*EF*) for Kumasi, Savè and Ile-Ife are shown in (d). Diamonds in (a), (b) and (c) indicate days with missing/incomplete radiosondes in Kumasi. Different phases of the monsoon are indicated according to Knippertz et al. (2017).**






**Figure 5: Daily resolved cloud-base fraction (a) and median (b) of clouds between 0 and 1000 m a.g.l. Distribution of cloud-base height occurrence during the whole 48 days period for the Kumasi (left) and Savè (right) sites (c). Cloud-base fraction and frequency distribution of cloud base height are calculated from ceilometer data. White horizontal lines in (a) indicate different phases of the monsoon according to Knippertz et al. (2017). Median diurnal cycles of the liquid water path, *LWP*, from microwave radiometer (d) and net longwave radiation from surface measurements (e). The shaded areas in corresponding colours represent interquartile ranges.**





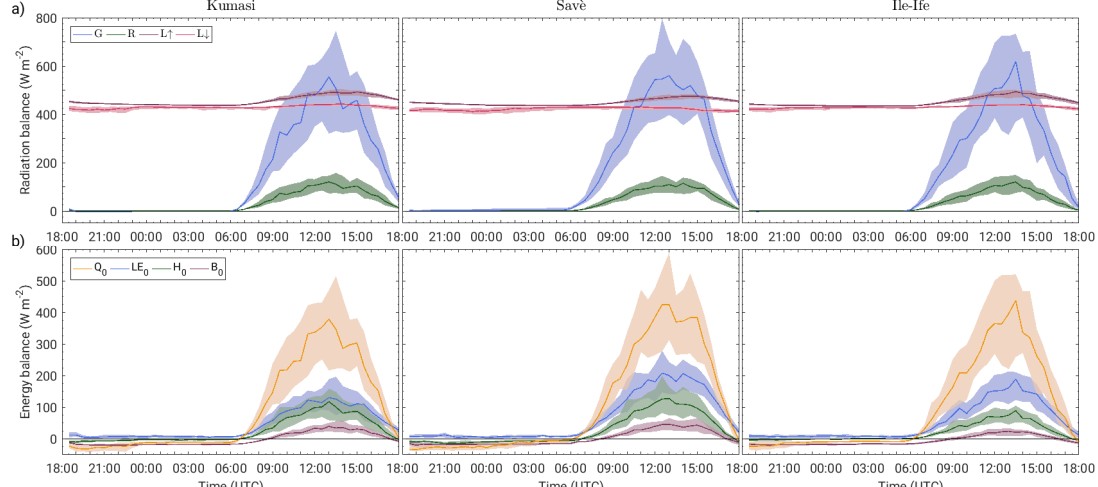

**Figure 6: (a) Median diurnal cycles of radiation balance components (shortwave downward, *G*, shortwave upward, *R*, longwave downward *L↓*, longwave upward, *L↑*, radiation) and the interquartile range (shaded) for Kumasi (left), Savè (middle) and Ile-Ife (right). b) Median diurnal cycles of energy balance components (net radiation, $Q_0$, sensible heat flux, $H_0$, latent heat flux, $LE_0$, and**

5     **soil heat flux, $B_0$) and the correspondingly coloured shaded areas represent interquartile ranges for Kumasi (left), Savè (middle) and Ile-Ife (right).**





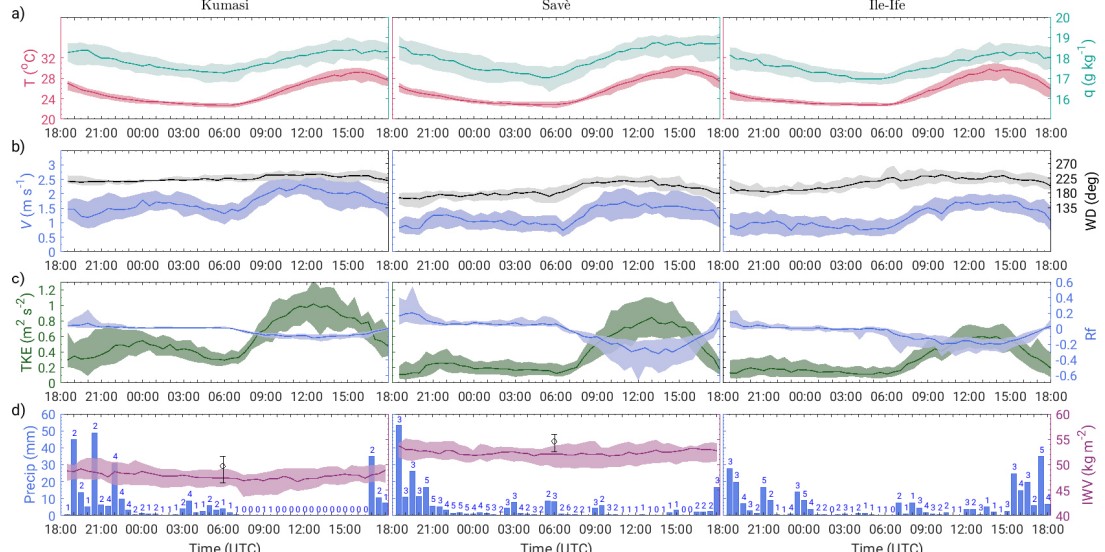

**Figure 7: a) Median diurnal cycles of specific humidity, *q*, temperature, *T*, and corresponding ranges between 25th and 75th percentile (shaded) for Kumasi (left), Savè (middle) and Ile-Ife (right). b) Medians of wind speed, *v*, and wind direction, *WD*, and the correspondingly coloured shaded areas represent interquartile ranges for Kumasi (left), Savè (middle) and Ile-Ife (right). c) Medians of turbulent kinetic energy, *TKE*, and flux Richardson number, *Rf*, with the median and the correspondingly coloured shaded areas represent interquartile ranges for Kumasi (left), Savè (middle) and Ile-Ife (right). d) Median diurnal cycles of integrated water vapour, *IWV*, and the correspondingly coloured shaded areas represent interquartile ranges for Kumasi (left) and Savè (middle). The additional observation at 0600 UTC indicates median value and interquartile ranges from radiosonde measurements. Half-hourly precipitation sums accumulated over the whole measurement period (numbers indicate the amount of days on which precipitation occurred).**





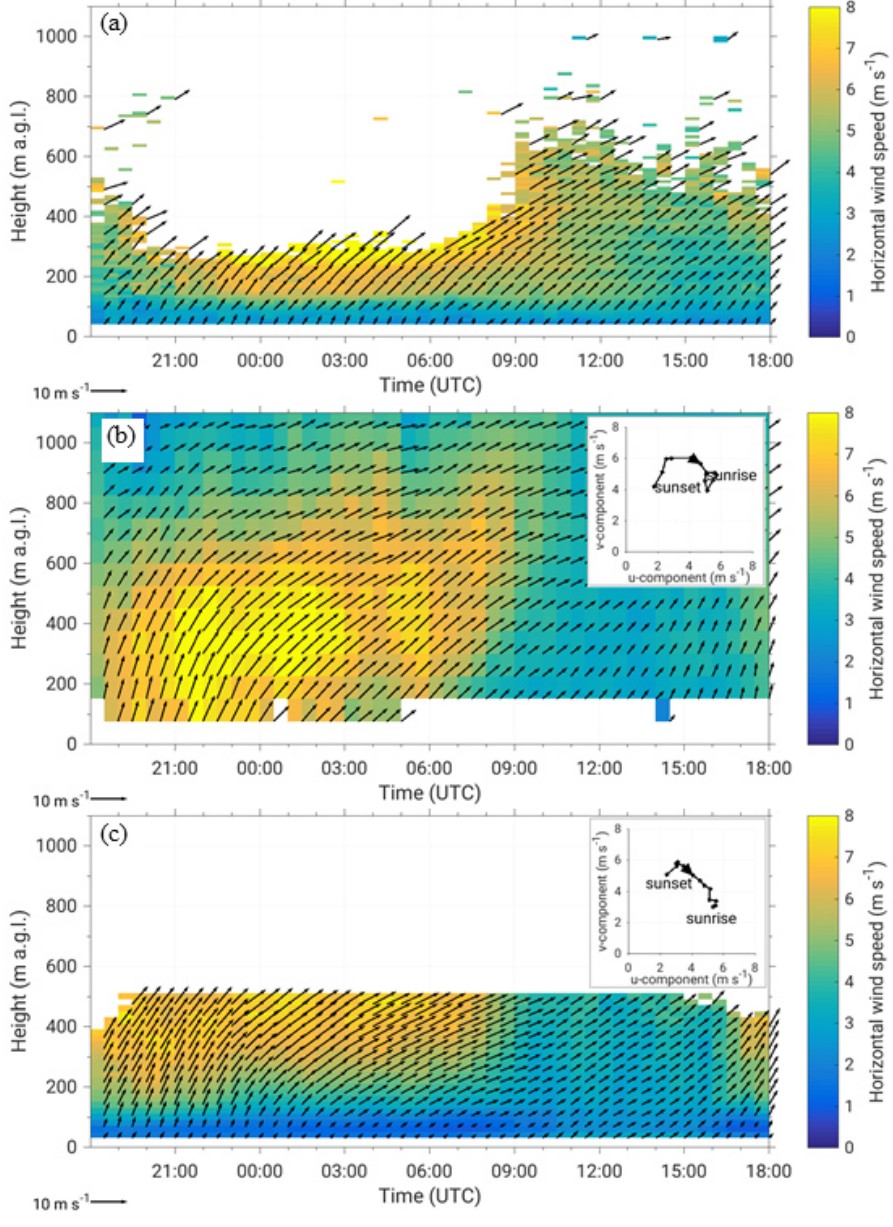

**Figure 8: Median diurnal horizontal wind speed (colour coded) and horizontal wind vectors (arrows) for Kumasi (a), Savè (b) and Ile-Ife (c). Additionally, in (b) and (c) the hodograph from 1800 UTC (approx. sunset) to 0600 UTC (approx. sunrise) is shown with the wind components averaged over one hour and between 200 and 600 m a.g.l.**