# Peer review of "An overview of the diurnal cycle of the atmospheric boundary layer during the West African monsoon season: Results from the 2016 observational campaign"

_Atmospheric Chemistry and Physics, 2017_

## Short Comment (SC1) · 20 Nov 2017

In this paper ceilometer measurements are used to determine the base heights of low level clouds. The authors use systems from two different manufacturers (Lufft, Campbell Scientific). To better understand the results presented in Section 4.1 and Figure 5 it is recommended to add some more details.

- What type of ceilometers has been used? I assume that a Lufft CHM15k Nimbus was deployed, but this should be confirmed. And a CS 135s?

[Figure]

- According to page 3 line 17 the cloud detection algorithms of the manufacturers have been applied. These algorithms might be conceptually different: one may define the cloud base height from the onset of the (strong) increase of the attenuated backscatter ($\beta^*$), from the maximum of $\beta^*$, from the inflection point of $\beta^*$ below the maximum, or something else. This could lead to a (small?) bias in the cloud base height retrievals when comparing results of different systems. Thus, a brief comment on how the software works should be added; this could help to interpret the results.

- It seems to me that the reason of the low cloud layer at approximately 100 m (page 8, line 9) is an artefact in the overlap correction function of the signals automatically applied by the Lufft software (in case of a Nimbus system). Such artefacts are frequently occurring – not only for the Lufft system. The applied overlap correction function is (to my knowledge) available from Lufft on request (at least for the most recent systems). Even if a broad discussion is beyond the goal of this paper this issue should be briefly mentioned/discussed.

  I do not know which kind of overlap correction is applied to the data of the Campbell Scientific system, but this certainly can be found out from the manufacturer.

A few papers that might be of help in this context (Kotthaus et al. discuss the overlap issue for a Vaisala ceilometer, but it is interesting as well), more are existing:

- Hervo, M., Poltera, Y., and Haefele, A.: An empirical method to correct for temperature-dependent variations in the overlap function of CHM15k ceilometers, Atmos. Meas. Tech., 9, 2947-2959, https://doi.org/10.5194/amt-9-2947-2016, 2016.

- Kotthaus, S., O'Connor, E., Münkel, C., Charlton-Perez, C., Haeffelin, M., Gabey, A. M., and Grimmond, C. S. B.: Recommendations for processing atmospheric

attenuated backscatter profiles from Vaisala CL31 ceilometers, Atmos. Meas. Tech., 9, 3769-3791, https://doi.org/10.5194/amt-9-3769-2016, 2016.

- Wiegner, M., Madonna, F., Binietoglou, I., Forkel, R., Gasteiger, J., Geiss, A., Pappalardo, G., Schäfer, K., and Thomas, W.: What is the benefit of ceilometers for aerosol remote sensing? An answer from EARLINET, Atmos. Meas. Tech., 7, 1979-1997, https://doi.org/10.5194/amt-7-1979-2014, 2014.

---

## Referee Comment (RC1) · A. Mekonnen (Referee) · 11 Dec 2017

I recommend publication after addressing comments of reviewers and concern from the open discussion.
* * *

---

## Referee Comment (RC2) · Anonymous Referee #1 · 20 Dec 2017

This paper describes initial analysis of ground-based measurements from a major field campaign in West Africa. In particular the paper focuses on measurements for characterising the occurrence of low-level clouds and the meteorological conditions in which they develop. The measurement systems and their geographical spread really help to fill a gap in our observational knowledge of this region and I'm sure will be of great value for many years to come. This paper provides a very well-written introduction to the dataset and the conditions over the course of the campaign. Aside from some minor grammatical suggestions, and taking into account the comment raised by Matthias

Wiegner on the use of different ceilometers, I consider this paper publishable with only very minor technical corrections in ACP. This paper (and the associated field campaign) is an extremely welcome addition to the body of knowledge on West African meteorology.

Suggested edits (to improve readability):

L8 "high-quality observational data in this region have been lacking"

L 11 "latent heat release, and. . ."

L21 "occurs frequently"

L22 "unclear what role. . . play in nocturnal. . ."?

L26 "These previous studies"?

L40 "in the lowest kilometres. . ."

L23 "to those which. . ."

L24 "are available from at least two of the supersites"

L:34 "for the Kumasi site. . ."

L7 "stratification, and to obtain"

L9 "lowest cloud layer which fulfils. . ."

L3 "with height above the ground"

L6 "...investigated in Section 4."

L36 "long-lived MCS"

L39 "during the monsoon onset phase"?

"(to approximately ... , and 80%..."

L22 "considerable variability from day to day with respect..."?

L40 "associated with large interquartile..."

L27 "Banta et al., 2006), and..."

L33 "To seek an explanation..."

L23 There is a missing verb here

L27 "stratified, being..."

L31 "every night, reaching maximum strength between..."

L5 "breaking up to form convective..."

Table 1 caption: "height in meters"

Figure 2 The arrows in the right-hand plots could be thicker

Figure 2 caption "(red dots) plus or minus their..."

[Figure]

---

## Author Comment (AC1) · 12 Jan 2018

M. Wiegner**, m.wiegner@lmu.de**,

In this paper ceilometer measurements are used to determine the base heights of low level clouds. The authors use systems from two different manufacturers (Lufft, Campbell Scientific). To better understand the results presented in Section 4.1 and Figure 5 it is recommended to add some more details.

- What type of ceilometers has been used? I assume that a Lufft CHM15k Nimbus was deployed, but this should be confirmed. And a CS 135s?

**Reply:** yes, a Lufft CHM15k and a Campbell Scientific CS135 were used. This information has been added to the text in section 2 (Pg. 4, line 13).

- According to page 3 line 17 the cloud detection algorithms of the manufacturers have been applied. These algorithms might be conceptually different: one may define the cloud base height from the onset of the (strong) increase of the attenuated backscatter ($\beta_*$), from the maximum of $\beta_*$, from the inflection point of $\beta_*$ below the maximum, or something else. This could lead to a (small?) bias in the cloud base height retrievals when comparing results of different systems. Thus, a brief comment on how the software works should be added; this could help to interpret the results.

**Reply:** We added a few sentences on how the algorithms of Lufft and Campbell work.

'Cloud-base height of CS135 is mainly based on an increasing slope of the extinction profile and an extinction threshold (see CS135 manual: https://s.campbellsci.com/documents/eu/manuals/cs135.pdf.), while cloud-base height of CHM15k is determined with a threshold method (personal communication). A comparison of both ceilometers is described in Madonna et al. (2015).' (Pg. 4, lines 15-18)

Cloud-base height calculation of CS135 is described in the CS135 manual (**Appendix C Cloud height calculation (applicable to OS8):**

*The scatter profile is inverted (using the Klett inversion technique) and an extinction profile is calculated.*
*Cloud base heights are identified using two criteria as follows (cloud is detected if either of them is met):*
*1. Criterion 1: likely cloud bases are estimated based on increasing slope of the extinction profile of at least 7m per bin (bin width is 5m) and an extinction threshold. This threshold is based on an extinction coefficient (EXCO) of 3, equivalent to a horizontal visibility (MOR) of 1,000m. This results in a number of possible cloud bases at different heights.*
*2. Criterion 2: horizontal visibility falls below an average of 4,800m over 300m starting at an altitude of 1000m.*

*To report a cloud layer above a lower one the scatter coefficient first has to fall below the extinction threshold used for cloud definitions (less a small hysteresis offset). The scatter profile must then again meet the criteria above.*

*To avoid many very narrow close layers being reported when they have little significance a minimum separation based on WMO reporting intervals is applied. The separation is +/- 30m below 1,500m and 300m above 1,500m. The lowest cloud height is used. If a thin cloud identified by Criterion 2 above is within +/-150m of a cloud identified by criterion 1 then the thin cloud is ignored.*

As only few detailed information on the cloud detection algorithm are available in the CHM15k manual (https://www.lufft.com/download/manual-lufft-chm15k-en/), we contacted Lufft for further information: the cloud detection algorithm applied for the CHM15k works with thresholds. In a first step, normalised backscatter signals are averaged in space and time. In a second step, clouds are detected when the signal exceeds a certain threshold.

- It seems to me that the reason of the low cloud layer at approximately 100 m (page 8, line 9) is an artefact in the overlap correction function of the signals automatically applied by the Lufft software (in case of a Nimbus system). Such artefacts are frequently occurring – not only for the Lufft system. The applied overlap correction function is (to my knowledge) available from Lufft on request (at least for the most recent systems). Even if a broad discussion is beyond the goal of this paper this issue should be briefly mentioned/discussed.

**Reply:** There is indeed a line with artificial high backscatter values at around 160 m which is related to the device specific overlap function. However, this line is not erroneously detected as clouds. Example backscatter plots and detected cloud bases for typical cases (one with cloud base height below 160 m and one with cloud base height above 160 m ) are shown in Fig. 1.

[Figure]

*Figure 1: Examples of backscatter measured by CHM15k and cloud bases (black dots) detected with the Lufft algorithm. The left figure shows an example for cloud bases below the line with artificial high backscatter at 160 m and the right figure an example for cloud bases above.*

I do not know which kind of overlap correction is applied to the data of the Campbell Scientific system, but this certainly can be found out from the manufacturer.

**Reply:** According to the manufacturer (personal communication), an empirical overlap function is applied to the CS135 data.

A few papers that might be of help in this context (Kotthaus et al. discuss the overlap issue for a Vaisala ceilometer, but it is interesting as well), more are existing:

- Hervo, M., Poltera, Y., and Haefele, A.: An empirical method to correct for temperature-dependent variations in the overlap function of CHM15k ceilometers, Atmos. Meas. Tech., 9, 2947-2959, https://doi.org/10.5194/amt-9-29472016, 2016.
- Kotthaus, S., O'Connor, E., Münkel, C., Charlton-Perez, C., Haeffelin, M., Gabey, A. M., and Grimmond, C. S. B.: Recommendations for processing atmospheric attenuated backscatter profiles from Vaisala CL31 ceilometers, Atmos. Meas. Tech., 9, 3769-3791, https://doi.org/10.5194/amt-9-3769-2016, 2016.
- Wiegner, M., Madonna, F., Binietoglou, I., Forkel, R., Gasteiger, J., Geiss, A., Pappalardo, G., Schäfer, K., and Thomas, W.: What is the benefit of ceilometers for aerosol remote sensing? An answer from EARLINET, Atmos. Meas. Tech., 7, 1979-1997, https://doi.org/10.5194/amt-7-1979-2014, 2014